# Longitudinal analysis of the risk factors for onset and change in tinnitus in the Busselton Healthy Ageing Study

**Denise Fuchten**[1,2]*, **Inge Stegeman**[1,2], **Yinan Mao**[1,2], **Robert H. Eikelboom**[3,4,5], **Michael L. Hunter**[6,7], **Adriana L. Smit**[1,2]

**1** Department of Otorhinolaryngology, Head and Neck Surgery, University Medical Centre Utrecht, Utrecht, The Netherlands, **2** University Medical Centre Utrecht Brain Centre, University Medical Centre Utrecht, Utrecht, The Netherlands, **3** Ear Science Institute Australia, Subiaco, Australia, **4** Ear Science Centre, The University of Western Australia, Nedlands, Australia, **5** Department of Speech Language Pathology and Audiology, University of Pretoria, Pretoria, South Africa, **6** Busselton Health Study Centre, Busselton Population Medical Research Institute, Busselton, Australia, **7** School of Population and Global Health, University of Western Australia, Crawley, Australia

* d.fuchten@umcutrecht.nl

## Abstract

### Introduction

Effective prevention and intervention strategies for tinnitus rely on identifying risk factors and understanding its progression over time. However, longitudinal data on these aspects are limited. This study therefore aimed to (1) assess the incidence of tinnitus and identify risk factors associated with tinnitus development, and (2) assess the impact of tinnitus and change in impact over time and identify factors associated with this change.

### Methods

Data from the Busselton Healthy Ageing Study, a population-based cohort of individuals born between 1946–1964, were used. Information on tinnitus presence and impact, general health, ear-related health and mental health was collected from 3863 participants through questionnaires and physical measurements at two time points with a six-year interval. Logistic regression analysis was used to examine risk factors for tinnitus development. Multinomial logistic regression analysis was used to examine factors associated with changes in impact.

### Results

The 6-year incidence of tinnitus was 12.1%. Statistically significant risk factors for developing tinnitus included male gender, higher BMI, larger waist circumference, fair subjective health, hearing loss, occupational noise exposure with occasional use of hearing protection, hyperacusis, migraine, and diagnosed anxiety disorder. Among

**Data availability statement:** The data that support the findings of this study are not publicly available. However, access to the data for research can be obtained by contacting the Busselton Population Medical Research Institute busseltonhealthstudy@health.wa.gov.au.

**Funding:** The author(s) received no specific funding for this work.

**Competing interests:** The authors have declared that no competing interests exist.

**Abbreviations:** ABI: Ankle Brachial Index; ADA: American Diabetes Association; AHA: American Heart Association; AUD: Australian Dollar; BHAS: Busselton Healthy Ageing Study; BMI: Body Mass Index; CI: Confidence Interval; CVD: Cardiovascular Disease; DASS: Depression, Anxiety and Stress Scale; dB: Decibel; DBP: Diastolic Blood Pressure; DM: Diabetes Mellitus; FPG: Fasting Plasma Glucose; HbA1c: Glycated Haemoglobin; IQR: Interquartile Range; OR: Odds Ratio; PAD: Peripheral artery disease; PHQ: Patient Health Questionnaire; PTA: Pure Tone Average; SBP: Systolic Blood Pressure; SD: Standard Deviation; TIA: Transient Ischemic Attack; TRQ: Tinnitus Reaction Questionnaire; WHO: World Health Organization.

participants with pre-existing tinnitus, 21.4% reported an increase in tinnitus impact over time, while 16.5% experienced a decrease. Changes in impact were influenced by general health and ear-related health factors.

## Conclusion

The high incidence of tinnitus and its notable impact on daily life emphasize the importance of gaining a better understanding of the broad range of identified risk factors for developing tinnitus and change in impact. The multifaceted nature of these factors, comprising hearing-related issues, general health conditions and psychological aspects, underscore the complexity of tinnitus etiology and impact. By gaining more insight into these factors, we can inform future research efforts aimed at developing targeted prevention and intervention strategies tailored to specific demographic groups.

## Introduction

Tinnitus, the perception of sound in the absence of an external auditory stimulus [1], is a heterogeneous condition in terms of etiology, perceptual experiences and associated suffering [2]. It is estimated that 14.4% of the adult population experiences tinnitus, therefore affecting more than 740 million people globally [3]. Its prevalence increases with age, making it more common among older populations [3,4]. This demographic trend raises concerns, as the rising life expectancy might indicate an even further increase in tinnitus cases. It is therefore imperative to identify risk factors for the development of tinnitus and understand its progression over time to develop and enhance prevention and intervention strategies.

Both otologic and non-otologic potential risk factors for tinnitus have previously been described, with hearing loss being the most reported risk factor [5–7]. However, most studies exploring these factors employ a cross-sectional design [5,6]. While this is valuable for identifying associations, it falls short in establishing a relationship over time. Most known information on longitudinally analyzed risk factors comes from large prospective cohort studies, such as the Blue Mountains Hearing Study [8] and the Epidemiology of Hearing Loss study [9,10]. In a meta-analysis examining tinnitus risk factors in such analytical observational studies by Biswas et al. (2023), the role of hearing loss in increasing the risk of tinnitus was confirmed [6]. Other factors, including otitis media, occupational noise exposure, ototoxic platinum exposure, temporo-mandibular joint disorder and depression, were also identified as risk factors, as was diabetes as a protective factor. However, these findings only resulted from pooling a relatively small number of studies, highlighting the lack of analytical observational studies [6].

While preventing tinnitus is a critical focus, it is equally important to consider the impact of tinnitus on those already affected. The distress caused by tinnitus can vary from person to person; while many report minimal to no impact of tinnitus, others are severely impaired in their daily lives due to this phantom sound. The psychoacoustic

characteristics of tinnitus, such as loudness and pitch, can only explain a small portion of the variation in experienced distress [11–13], while psychological factors and coping mechanisms play a crucial role in the severity of distress experienced [14,15]. Additionally, the impact of tinnitus can change over time. Evidence suggests that a significant proportion of individuals with tinnitus experience a reduction in the experienced distress over time [16,17], possibly due to habituation mechanisms [18]. However, the change in impact can be highly variable across individuals [17] and data on the long-term course of tinnitus is scarce [19]. Understanding these patterns and the factors influencing them is vital for effective long-term management of tinnitus.

The aim of the present study is therefore twofold: (1) to assess the incidence of tinnitus and identify risk factors associated with the development of tinnitus, and (2) to assess the impact of tinnitus and change in impact over time and identify factors associated with this change. Recognizing the importance of identifying these factors within the general population, and considering the role of aging in tinnitus, this study was conducted using longitudinal data from the Busselton Healthy Aging Study, a study involving middle-aged community dwelling participants.

## Methods

### Study design and population

This study was conducted with data from the Busselton Healthy Ageing Study (BHAS), a longitudinal study of community dwelling adults born between 1946 and 1964 residing in the city of Busselton, Western Australia. All eligible adults registered on the compulsory national electoral roll, totaling 6690 individuals, were invited to participate. Invitations were sent out in randomized order through sequential 10% sample draws. Participants were invited via a letter of introduction, followed by a phone call to invite them to the test center for a four-hour appointment to complete health-related questionnaires and a range of physical and cognitive assessments.

Phase I of the study was conducted from 17 May 2010–4 December 2015, in which 5107 participants were included. The follow-up phase II was conducted from 14 March 2016–24 January 2022, approximately six years after the baseline measurement, and comprised 3888 participants (~ 76% of initial participants). The full protocol of the BHAS has been published previously [20], and additional documentation provides the complete inventory of all variables collected during both phases [21]. The study received ethics approval from The University of Western Australia Human Research Ethics Committee (Number RA/4/1/2203). All participants gave their informed written consent before inclusion in the study. This study is reported according to the STROBE statement.

### Measures

Data for this study were obtained from a self-administered health and risk factor questionnaire, alongside comprehensive physical assessments. These assessments included measurements of body composition, fasting blood samples, blood pressure measurements and audiometry.

### Demographics

Demographic data included age, gender, marital status, highest level of educational attainment (at phase I) and annual household income in Australian dollars (AUD).

### Tinnitus

The presence of tinnitus was determined by the question "Do you experience tinnitus (sound in your ears and head) for longer than 5 minutes, which does not have an obvious cause?". If participants answered affirmatively, the impact of tinnitus was assessed with the question "How much does tinnitus affect your daily life and activities?" with response options "not at all", "occasionally", "frequently" or "constantly".

To gain insight into these impact categories, we also included the scores on the Tinnitus Reaction Questionnaire (TRQ), administered during phase II, to examine its alignment to these categories. The TRQ is a 26-item questionnaire designed to assess the psychological distress associated with tinnitus [22]. Items on this questionnaire are scored on a five-point scale ranging from 0 (not at all) to 4 (almost all of the time), and the total score ranges from 0–104, with higher scores indicating more tinnitus related psychological distress.

**General health**

Standing height, waist circumference and weight were measured using standard anthropometric techniques. Body Mass Index (BMI) was calculated (kg/m²) and analyzed as a continuous outcome. For descriptive purposes, BMI was also categorized as follows: underweight (<18.5 kg/m2), healthy weight (18.5–24.9 kg/m2), overweight (25.0–29.9 kg/m2) and obese (>30 kg/m2).

Self-reported general health was assessed by the question "In general, would you say your health is" with response options "excellent", "very good", "good", "fair" or "poor".

Cumulative smoking behaviour in pack years was determined from the amount (number of cigarettes participants currently smoke or previously smoked per day) and duration (years of smoking based on age of starting and stopping) of smoking. Pack years were calculated as the equivalent of smoking 20 cigarettes per day for one year and reported as a continuous variable. For descriptive purposes, pack years of smoking was also categorized as follows: never smokers (0), light smokers (0.1–20), moderate smokers (20.1–40) and heavy smokers (> 40).

Cardiovascular disease status (yes/no) of a participant was determined based on the presence of heart disease, a history of stroke, or peripheral artery disease (PAD). Criteria for heart disease included a positive history of angina or myocardial infarction, the presence of an implant or cardiac pacemaker, and having undergone coronary angioplasty, stent placement, or coronary bypass surgery. A history of stroke was defined by a positive history of a transient ischemic attack (TIA) or stroke, or by having undergone carotid surgery. PAD was indicated by an ankle-brachial index (ABI) of <0.9 in either leg, or, if ABI data were missing, by a positive history of claudication.

Blood pressure was measured with the participant in a supine position after five minutes of rest. Measurements included systolic blood pressure (SBP) and diastolic blood pressure (DBP) and were categorized according to the classification of the American Heart Association: normal (SBP < 120 mm Hg and DBP < 80 mm Hg), elevated (SBP 120–129 mm Hg and DBP < 80 mm Hg), hypertension stage 1 (SBP 130–139 mm Hg or DBP 80–89 mm Hg), and hypertension stage 2 (SBP ≥ 140 mm Hg or DBP ≥ 90 mm Hg) [23].

To determine diabetes status, participants were asked whether they had received a doctor's diagnosis of diabetes. Additionally, a fasting venous blood sample was collected for analysis of general chemistry of plasma (Fasting Plasma Glucose; FPG) and whole blood (glycated hemoglobin; HbA1c). Classification followed the criteria set by the American Diabetes Association (ADA) 2024 guidelines: pre-diabetes (HbA1C 39–47 mmol/mol or FPG 5.6–6.9 mmol/L), and diabetes (HbA1C > ≥48 mmol/mol or FPG > 7.0 mmol/L) [24].

Doctor-diagnosed medical history of cancer, migraine, and head injury involving loss of consciousness and hospitalization for ≥ 1 day were based on participants self-reporting.

**Ear-related health**

Doctor-diagnosed medical history of Meniere's disease and chronic ear infection were based on participants self-reporting. The presence of hyperacusis was determined based on the question "Do you consider yourself sensitive or intolerant to everyday sounds?"

Occupational noise exposure was determined using the question "Have you worked in a place where it was so noisy that you had to raise your voice to be heard by others?". If answered affirmatively, participants were asked "Did you wear hearing protection?" with answer options "never", "occasionally", "frequently" or "always". Based on these questions,

participants were categorized in one of the following groups: quiet workplace, noisy workplace – always protected, noisy workplace – frequently protected, noisy workplace – occasionally protected or noisy workplace – never protected.

Pure tone air conduction thresholds were determined using automated audiometry in accordance with the Hughson-Westlake methods. Participants were asked to respond with "Yes" or "No" after the presentation of every stimulus using a touch screen. Air conduction thresholds were recorded at 250, 500, 1000, 2000, 4000 and 8000 Hz. The four-frequency pure tone average (PTA) was provided as the mean of the air conduction hearing levels for 500, 1000, 2000, and 4000 Hz in both the best (lowest average) and worst (highest average) ear. Subsequently, for descriptive purposes, hearing levels were categorized in scales of impairment based on the WHO classification as follows: no impairment (< 20 dB HL), mild (20–34 dB HL), moderate (35–49 dB HL), moderately severe (50–64 dB HL), severe (65-79 dB HL), profound (80–94 dB HL) [25].

### Mental health

The Patient Health Questionnaire-9 (PHQ-9), the Depression, Anxiety and Stress Scale-21 (DASS-21), and the questions "Have you ever been told by a doctor that you have depression?" and "Have you ever been told by a doctor that you have an anxiety disorder (including Post Traumatic Stress Disorder)?" were used to assess depressive, anxiety and stress symptoms, and to assess the diagnoses of depression and anxiety.

The PHQ-9 is the nine-item depression module of the PHQ [26]. The items are based on the DSM-IV criteria for depressive disorders and can be scored on a four-point scale ranging from 0 (not at all) to 3 (nearly every day). The total score ranges from 0–27, where a higher score indicates more (severe) depressive symptoms. For descriptive purposes, this score was categorized as follows: minimal (1–4), mild (5–9), moderate (10–14), moderately severe (15–19), severe (20–27). Additionally, a scoring method based on the DSM-IV criteria was used to categorize answers into suspected major depressive disorder and suspected other depressive disorders [27].

The DASS-21 measures depression, anxiety, and stress across 21 items divided into three subscales with seven items, each scored from 0 (did not apply to me at all) to 3 (applied to me very much, most of the time) [28]. Total subscale scores can range from 0–42. The total score is calculated by summing the scores and then multiplying by two, with higher scores indicating more severe symptoms. The scores were also categorized for descriptive purposes. For the depression subscale: normal (0–9), mild (10–13), moderate (14–20), severe (21–27), extremely severe (28+). For the anxiety subscale: normal (0–7), mild (8–9), moderate (10–14), severe (15–19), extremely severe (20+). For the stress subscale: normal (0–14), mild (15–18), moderate (19–25), severe (26–33), extremely severe (34 + points) [28].

### Statistical analyses

All statistical analysis were performed using IBM SPSS statistics 29.0.1 and R version 4.3.2, with additional packages 'tidyverse' for data wrangling, 'haven' and 'labelled' for SPSS file import, 'mice' for missing data imputation, 'nnet' for multinomial logistic regression analysis and 'grid' for plotting graphs. A value of $p < .05$ was considered statistically significant.

All variables in this study were checked for missing data. Multivariate imputation by chained equations (MICE) was used to address these missing values, with predictive mean matching applied iteratively for each variable. The imputed dataset was saved and used for analysis.

Demographic variables and measures of tinnitus, general health, ear-related health and mental health were reported using means with standard deviations for continuous variables and using counts with percentages for categorical variables.

Incidence of tinnitus over the two phases was calculated and logistic regression was applied to examine potential risk factors, adjusted for age and gender, for the development of tinnitus. For each potential risk factor, the corresponding odds ratio and the associated 95% confidence interval were reported.

Change in tinnitus impact between the two phases was categorized as reduced (less impact in phase II than phase I), increased (more impact in phase II than phase I) and same impact (no difference between the two phases). Multinomial logistic regression, adjusted for age and gender, examined factors associated with changes in tinnitus impact, using the same impact category as the reference. Odds ratios and 95% confidence intervals were reported for both reduced and increased impact categories.

## Results

### Characteristics of study sample

3888 participants completed both phase I and phase II measurements. Of those, 25 participants (0.6%) could not be matched across the two phases due to missing IDs. Consequently, 3863 participants were included in the analysis. Among these participants, 239 (6.2%) had missing values across one or multiple variables, which were therefore imputed.

The mean age of the study population was 58.0 (SD = 5.7) years at phase I and 64.2 (SD = 5.4) years at phase II, and 2134 (55.2%) participants were female (Table 1).

### Tinnitus prevalence and incidence

881 of the 3863 participants (22.8%) experienced tinnitus during phase I (Table 2). This number increased to 1084 (28.1%) during phase II. Of the 2982 participants who did not report tinnitus at phase I, 361 participants reported experiencing tinnitus at phase II, indicating new-onset tinnitus over the six-year period (Table 2). The estimated six-year incidence rate of tinnitus onset was 0.12 (95% CI [0.11–0.13]). This corresponds to an incidence of 2017.66 per 100,000 person-years, calculated using the unrounded incidence estimate.

### Risk factors for the development of tinnitus

The outcomes of the logistic regression analysis for all included variables are displayed in Table 3. Some of the notable results of the analysis are highlighted in this section.

In the demographic domain, sex was identified as a significant risk factor, with males being more likely to develop tinnitus compared to females (OR = 1.42, 95% CI [1.14–1.77], $p < 0.005$). Age, however, did not affect the odds ratio (OR = 1.00, 95% CI [0.98–1.02], $p = 0.84$). Although not statistically significant, lower income levels (less than $20,000: OR = 1.67, 95% CI [0.99–2.83], $p = 0.06$; $20,001–$40,000: OR = 1.43, 95% CI [1.00–2.06], $p = 0.05$) were associated with higher odds of developing tinnitus when compared to the highest income category. Highest educational attainment was not identified as a significant risk factor.

When analyzing the variables within the general health domain, a fair subjective general health compared to a good health was significantly associated with a higher likelihood of developing tinnitus (OR = 1.55, 95% CI [1.00–2.39], $p = 0.049$), but not poor subjective general health (OR = 1.03, 95% CI [0.23–4.59], $p = 0.97$). Cardiovascular disease (OR = 1.29, 95% CI [0.79–2.12], $p = 0.31$), pre-diabetes (OR = 0.84, 95% CI [0.66–1.06], $p = 0.14$) and diabetes (OR = 1.14, 95% CI [0.75–1.75], $p = 0.53$) were not significantly associated with incident tinnitus, and neither were an elevated blood pressure (OR = 1.03, 95% CI [0.72–1.47], $p = 0.87$), hypertension I (OR = 1.31, 95% CI [0.97–1.77], $p = 0.07$), and hypertension II (OR = 1.45, 95% CI [0.98–2.13], $p = 0.06$). The presence of migraine, however, did significantly increase the likelihood of developing tinnitus with an odds ratio of 1.44 (95% CI [1.08–1.91], $p = 0.01$).

Within the ear-related health domain, hyperacusis was identified as a significant risk factor for tinnitus development (OR = 1.40, 95% CI [1.01–1.94], $p = 0.04$), as was occupational noise exposure for those only occasionally protected compared to those working in a quiet workplace (OR = 1.48, 95% CI [1.05–2.08], $p = 0.02$). Other noise exposure categories were non-significant: always protected (OR = 1.30, 95% CI [0.76–2.22], $p = 0.34$), frequently protected (OR = 1.15, 95% CI [0.73–1.80], $p = 0.55$), and never protected (OR = 1.03, 95% CI [0.71–1.50], $p = 0.86$).

 

**Table 1. Characteristics of the study sample during phase I and phase II.**

| | Phase I | | | Phase II | | |
|---|---|---|---|---|---|---|
| | Tinnitus yes n=881 | Tinnitus no n=2982 | Total n=3863 | Tinnitus yes n=1084 | Tinnitus no n=2779 | Total n=3863 |
| **Sex** | | | | | | |
| Female | 371 (42.1%) | 1763 (59.1%) | 2134 (55.2%) | 475 (43.8%) | 1659 (59.7%) | 2134 (55.2%) |
| Male | 510 (57.9%) | 1219 (40.9%) | 1729 (44.8%) | 609 (56.2%) | 1120 (40.3%) | 1729 (44.8%) |
| **Age (years)** | | | | | | |
| Mean (SD) | 59.10 (5.63) | 57.62 (5.70) | 57.96 (5.72) | 64.88 (5.27) | 63.89 (5.40) | 64.17 (5.38) |
| (45,50] | 67 (7.6%) | 333 (11.2%) | 400 (10.4%) | 0 (0.0%) | 0 (0.0%) | 0 (0.0%) |
| (50,55] | 163 (18.5%) | 763 (25.6%) | 926 (24.0%) | 25 (2.3%) | 117 (4.2%) | 142 (3.7%) |
| (55,60] | 215 (24.4%) | 786 (26.4%) | 1001 (25.9%) | 208 (19.2%) | 652 (23.5%) | 860 (22.3%) |
| (60,65] | 303 (34.4%) | 762 (25.6%) | 1065 (27.6%) | 297 (27.4%) | 827 (29.8%) | 1124 (29.1%) |
| (65,70] | 133 (15.1%) | 338 (11.3%) | 471 (12.2%) | 343 (31.6%) | 761 (27.4%) | 1104 (28.6%) |
| (70,80) | 0 (0.0%) | 0 (0.0%) | 0 (0.0%) | 211 (19.5%) | 422 (15.2%) | 633 (16.4%) |
| **Education** | | | | | | |
| Primary school | 7 (0.8%) | 30 (1.0%) | 37 (1.0%) | 8 (0.7%) | 29 (1.0%) | 37 (1.0%) |
| Secondary school | 429 (48.7%) | 1394 (46.7%) | 1823 (47.2%) | 515 (47.5%) | 1308 (47.1%) | 1823 (47.2%) |
| University | 154 (17.5%) | 642 (21.5%) | 796 (20.6%) | 200 (18.5%) | 596 (21.4%) | 796 (20.6%) |
| Other | 291 (33.0%) | 915 (30.7%) | 1206 (31.2%) | 361 (33.3%) | 845 (30.4%) | 1206 (31.2%) |
| No school | 0 (0.0%) | 1 (0.0%) | 1 (0.0%) | 0 (0.0%) | 1 (0.0%) | 1 (0.0%) |
| **Income** | | | | | | |
| Less than $20,000 | 57 (6.5%) | 143 (4.8%) | 200 (5.2%) | 67 (6.2%) | 127 (4.6%) | 194 (5.0%) |
| $20,001 to $40,000 | 137 (15.6%) | 462 (15.5%) | 599 (15.5%) | 189 (17.4%) | 475 (17.1%) | 664 (17.2%) |
| $40,001 to $60,000 | 154 (17.5%) | 500 (16.8%) | 654 (16.9%) | 161 (14.9%) | 434 (15.6%) | 595 (15.4%) |
| $60,001 to $80,000 | 131 (14.9%) | 388 (13.0%) | 519 (13.4%) | 124 (11.4%) | 334 (12.0%) | 458 (11.9%) |
| $80,001 to $100,000 | 105 (11.9%) | 363 (12.8%) | 488 (12.6%) | 131 (12.1%) | 322 (11.6%) | 453 (11.7%) |
| More than $100,000 | 193 (21.9%) | 736 (24.7%) | 929 (24.0%) | 187 (17.3%) | 567 (20.4%) | 754 (19.5%) |
| Unknown | 104 (11.8%) | 370 (12.4%) | 474 (12.3%) | 225 (20.8%) | 520 (18.7%) | 745 (19.3%) |
| **Marital status** | | | | | | |
| Single | 41 (4.7%) | 152 (5.1%) | 157 (4.1%) | 64 (5.9%) | 137 (4.9%) | 201 (5.2%) |
| Married | 672 (76.3%) | 2223 (74.5%) | 2962 (76.7%) | 816 (75.3%) | 2068 (74.4%) | 2884 (74.7%) |
| Widowed | 20 (2.3%) | 127 (4.3%) | 95 (2.5%) | 41 (3.8%) | 121 (4.4%) | 162 (4.2%) |
| Divorced | 62 (7.0%) | 233 (7.8%) | 274 (7.1%) | 75 (6.9%) | 217 (7.8%) | 292 (7.6%) |
| Separated | 20 (2.3%) | 68 (2.3%) | 90 (2.3%) | 18 (1.7%) | 66 (2.4%) | 84 (2.2%) |
| De facto | 66 (7.5%) | 179 (6.0%) | 285 (7.4%) | 70 (6.5%) | 170 (6.1%) | 240 (6.2%) |
| **Waist (cm)** | | | | | | |
| Mean (SD) | 94.21 (13.12) | 91.02 (13.35) | 91.75 (13.36) | 95.97 (13.26) | 92.80 (13.73) | 93.69 (13.67) |
| **BMI** | | | | | | |
| Mean (SD) | 28.43 (4.90) | 27.92 (4.85) | 28.04 (4.86) | 28.78 (4.94) | 28.16 (5.02) | 28.33 (5.00) |
| Underweight | 2 (0.2%) | 14 (0.5%) | 16 (0.4%) | 3 (0.3%) | 11 (0.4%) | 14 (0.4%) |
| Normal | 202 (22.9%) | 843 (28.3%) | 1045 (27.1%) | 236 (21.8%) | 770 (27.7%) | 1006 (26.0%) |
| Overweight | 413 (46.9%) | 1277 (42.8%) | 1690 (43.7%) | 476 (43.9%) | 1125 (40.5%) | 1601 (41.4%) |
| Obese | 264 (30.0%) | 848 (28.4%) | 1112 (28.8%) | 369 (34.0%) | 873 (31.4%) | 1242 (32.2%) |

Lastly, within the mental health domain, clinician-diagnosed anxiety showed a statistically significant odds ratio of 1.63 (95% CI [1.10–2.41], p=0.01), while clinician-diagnosed depression did not show a significant result (OR = 1.19, 95% CI [0.92–1.53], p=0.18).

**Table 2. Number of people reporting tinnitus presence ('yes') or absence ('no') during phase I and II.**

| | | Phase II | | |
| --- | --- | --- | --- | --- |
| | | Yes | No | |
| Phase I | Yes | 723 (18.7%) | 158 (4.1%) | 881 (22.8%) |
| | No | 361 (9.3%) | 2621 (67.8%) | 2982 (77.2%) |
| | | 1084 (28.1%) | 2779 (71.9%) | |

## Tinnitus impact and change in impact

For participants with tinnitus, the impact on daily life and activities is detailed in Table 4. During phase 1, 366 (41.5%) participants with tinnitus reported that tinnitus at least occasionally affected their daily life and activities, with 73 (8.3%) participants indicating that it frequently or constantly affected them. During phase II these numbers were 449 (41.4%) and 95 (8.7%) respectively.

Of the 1084 participants with tinnitus during phase II, 1079 (99.5%) participants also completed the TRQ. Participants who indicated that tinnitus did not affect their daily life or activities, had a mean TRQ score of 1.73. The TRQ score increased when the impact increased to occasionally, frequently and constantly (mean scores respectively 7.93, 19.46, 30.89). Table 5 shows the TRQ descriptives per tinnitus impact category.

As shown in Table 2, 723 of the 3863 participants (18.7%) experienced tinnitus in both phase I and II. Of these participants, 450 (62.2%) did not report a change in the impact tinnitus had on their daily life and activities. 155 participants (21.4%) reported an increased impact of tinnitus, while 119 (16.5%) participants reported the impact of tinnitus decreased. Table 6 shows the (change in) impact of all participants who had tinnitus during phase I and II.

## Factors associated with change in tinnitus impact

The outcomes of the multinomial logistic regression analysis for all included variables are displayed in Table 7. Some of the notable results of the analysis are highlighted in this section.

When analyzing factors contributing to a reduction in tinnitus impact, both age (OR = 1.01, 95% CI [0.97–1.05], p = 0.78) and gender (OR = 1.29, 95% CI [0.85–0.96], p = 0.24) did not play a significant role. An income of less than $20,000, however, significantly reduced the impact participants experienced over time, with an odds ratio of 2.77 (95% CI [1.21–6.31], p = 0.02), while the other income levels were not associated with tinnitus impact decreasing. Highest educational attainment was also not identified as a significant factor.

Within the domain of general health, an excellent subjective general health compared to a good health, showed a significantly reduced likelihood of tinnitus impact decreasing over time (OR = 0.30, 95% CI = 0.12–0.73, p = 0.01). Conversely, cardiovascular disease (OR = 3.41, 95% CI [1.78–6.54], p < 0.005) and diabetes (OR = 2.25, 95% CI [1.16–4.36], p = 0.02) both were significantly associated with tinnitus impact decreasing over time. An elevated blood pressure (OR = 0.74, 95% CI [0.39–1.38], p = 0.34), hypertension I (OR = 0.61, 95% CI [0.36–1.05], p = 0.07) and hypertension II (OR = 0.61, 95% CI [0.30–1.21], p = 0.15) were not significant.

In the domain of ear-related health factors, the presence of hyperacusis did not significantly decrease tinnitus impact (OR = 1.38, 95% CI [0.88–2.15], p = 0.16). Occupational noise exposure, however, was a significant factor; for those occasionally protected compared to those in a quiet workplace the likelihood of tinnitus reduction increased (OR = 1.91, 95% CI [1.02–3.57], p = 0.049). Other noise exposure categories were non-significant: always protected (OR = 2.12, 95% CI [0.81–5.55], p = 0.13), frequently protected (OR = 1.15, 95% CI [0.53–2.50], p = 0.72), and never protected (OR = 1.69, 95% CI [0.94–3.04], p = 0.08).

Lastly, when looking at mental health factors, both doctor diagnosed depression (OR = 1.29, 95% CI [0.84–1.99], p = 0.24) and doctor diagnosed anxiety (OR = 1.66, 95% CI [0.86–3.21], p = 0.13) were not significantly associated with a decrease in tinnitus impact over time.

**Table 3. Outcomes of the logistic regression analysis assessing the risk factors for developing tinnitus.**

| | Tinnitus at phase II (n = 361) | No Tinnitus at phase II (n = 2621) | Unadjusted analysis | | Adjusted analysis | |
| --- | --- | --- | --- | --- | --- | --- |
| | N (%) | N (%) | OR (95% CI) | *p* | OR (95% CI) | *p* |
| **Sex** | | | | | | |
| Female (ref) | 186 (51.5%) | 1577 (60.2%) | | | | |
| Male | 175 (48.5%) | 1044 (39.8%) | 1.42 (1.14-1.77) | <0.005 | 1.42 (1.14-1.77) | <0.005 |
| **Age in years, mean (SD)** | 57.58 (5.57) | 57.62 (5.72) | 1.00 (0.98-1.02) | 0.88 | 1.00 (0.98-1.02) | 0.84 |
| (45,50] | 38 (10.5%) | 295 (11.3%) | | | | |
| (50,55] | 95 (26.3%) | 668 (25.5%) | | | | |
| (55,60] | 108 (29.9%) | 678 (25.9%) | | | | |
| (60,65] | 76 (21.1%) | 686 (26.2%) | | | | |
| (65,70] | 44 (12.2%) | 294 (11.2%) | | | | |
| **Marital status** | | | | | | |
| Single | 16 (4.4%) | 100 (3.8%) | 1.13 (0.65-1.94) | 0.67 | 1.11 (0.65-1.92) | 0.70 |
| Married (ref) | 285 (78.9%) | 2005 (76.5%) | | | | |
| Widowed | 3 (0.8%) | 72 (2.7%) | 0.29 (0.09-0.94) | 0.04 | 0.32 (0.10-1.03) | 0.06 |
| Divorced | 26 (7.2%) | 186 (7.1%) | 0.99 (0.64-1.52) | 0.96 | 1.05 (0.68-1.61) | 0.83 |
| Separated | 9 (2.5%) | 61 (2.3%) | 1.04 (0.51-2.11) | 0.92 | 1.09 (0.53-2.22) | 0.81 |
| De facto | 22 (8.1%) | 197 (7.5%) | 0.78 (0.49-1.24) | 0.29 | 0.77 (0.49-1.22) | 0.27 |
| **Income** | | | | | | |
| Les than $20,000 | 22 (6.1%) | 121 (4.6%) | 1.43 (0.86-2.37) | 0.17 | 1.67 (0.99-2.83) | 0.06 |
| $20,001-$40,000 | 65 (18.0%) | 397 (15.1%) | 1.29 (0.91-1.82) | 0.16 | 1.43 (1.00-2.06) | 0.05 |
| $40,001-$60,000 | 58 (16.1%) | 442 (16.9%) | 1.03 (0.72-1.47) | 0.87 | 1.11 (0.77-1.60) | 0.59 |
| $60,001-$80,000 | 40 (11.1%) | 348 (13.3%) | 0.90 (0.61-1.35) | 0.62 | 0.94 (0.63-1.41) | 0.77 |
| $80,000-$100,000 | 52 (14.4%) | 331 (12.6%) | 1.23 (0.85-1.79) | 0.27 | 1.27 (0.88-1.85) | 0.20 |
| More than $100,000 (ref) | 84 (23.3%) | 652 (24.9%) | | | | |
| Unknown | 40 (11.1%) | 330 (12.6%) | 0.98 (0.66-1.45) | 0.90 | 1.08 (0.72-1.61) | 0.73 |
| **Highest education** | | | | | | |
| Did not go to school | 0 (0.0%) | 1 (0.0%) | 2.58 (9.66-6.92) | 0.97 | 0.00 (0.00-8.27) | 0.97 |
| Primary school | 3 (0.8%) | 27 (1.0%) | 0.82 (0.25-2.74) | 0.75 | 0.81 (0.24-2.71) | 0.73 |
| Secondary school (ref) | 165 (45.7%) | 1229 (46.9%) | | | | |
| University | 75 (20.8%) | 567 (21.6%) | 1.10 (0.85-1.41) | 0.47 | 0.98 (0.73-1.32) | 0.90 |
| Other educational institute | 118 (32.7%) | 797 (30.4%) | 0.97 (0.72-1.30) | 0.83 | 1.14 (0.88-1.47) | 0.31 |
| **BMI, mean (SD)** | 28.47 (4.93) | 27.85 (4.83) | 1.03 (1.00-1.05) | 0.02 | 1.02 (1.00-1.05) | 0.04 |
| Underweight | 1 (0.3%) | 13 (0.5%) | | | | |
| Normal | 87 (24.1%) | 756 (28.8%) | | | | |
| Overweight | 163 (45.2%) | 1114 (42.5%) | | | | |
| Obese | 110 (30.5%) | 738 (28.2%) | | | | |
| **Waist in cm, mean (SD)** | 93.14 (12.95) | 90.73 (13.38) | 1.01 (1.01-1.02) | <0.005 | 1.01 (1.00-1.02) | 0.03 |
| **General health** | | | | | | |
| Poor | 2 (0.6%) | 14 (0.5%) | 1.14 (0.26-5.08) | 0.86 | 1.03 (0.23-4.59) | 0.97 |
| Fair | 30 (8.3%) | 153 (5.8%) | 1.57 (1.01-2.42) | 0.04 | 1.55 (1.00-2.39) | 0.049 |
| Good (ref) | 121 (33.5%) | 967 (36.9%) | | | | |
| Very good | 154 (42.7%) | 1085 (41.4%) | 1.13 (0.88-1.46) | 0.33 | 1.16 (0.90-1.50) | 0.25 |
| Excellent | 54 (15.0%) | 402 (15.3%) | 1.07 (0.76-1.51) | 0.68 | 1.11 (0.79-1.56) | 0.56 |
| **Smoking in pack years, mean (SD)** | 8.04 (13.77) | 7.54 (14.29) | 1.00 (0.99-1.01) | 0.53 | 1.00 (0.99-1.01) | 0.93 |
| Never smokers | 179 (49.6%) | 1348 (51.4%) | | | | |

*(Continued)*

**Table 3.** (Continued)

| | Tinnitus at phase II (n = 361) | No Tinnitus at phase II (n = 2621) | Unadjusted analysis | | Adjusted analysis | |
|---|---|---|---|---|---|---|
| | N (%) | N (%) | OR (95% CI) | p | OR (95% CI) | p |
| Light smokers | 131 (36.3%) | 864 (33.0%) | | | | |
| Moderate smokers | 34 (9.4%) | 234 (8.9%) | | | | |
| Heavy smokers | 17 (4.7%) | 175 (6.7%) | | | | |
| **CVD** | | | | | | |
| No (ref) | 341 (94.5%) | 2512 (95.8%) | | | | |
| Yes | 20 (5.5%) | 109 (4.2%) | 1.35 (0.83-2.21) | 0.23 | 1.29 (0.79-2.12) | 0.31 |
| **Blood pressure** | | | | | | |
| Normal (ref) | 75 (20.8%) | 674 (25.7%) | | | | |
| Elevated | 64 (17.7%) | 539 (20.6%) | 1.06 (0.74-1.50) | 0.76 | 1.03 (0.72-1.47) | 0.87 |
| Hypertension I | 169 (46.8%) | 1090 (41.6%) | 1.38 (1.03-1.84) | 0.03 | 1.31 (0.97-1.77) | 0.07 |
| Hypertension II | 53 (14.7%) | 318 (12.1%) | 1.50 (1.03-2.18) | 0.04 | 1.45 (0.98-2.13) | 0.06 |
| **Diabetes** | | | | | | |
| No (ref) | 191 (52.9%) | 1306 (49.8%) | | | | |
| Pre-DM | 141 (39.1%) | 1143 (43.6%) | 0.84 (0.67-1.06) | 0.15 | 0.84 (0.66-1.06) | 0.14 |
| DM | 29 (8.0%) | 172 (6.6%) | 1.15 (0.76-1.76) | 0.51 | 1.14 (0.75-1.75) | 0.53 |
| **Cancer** | | | | | | |
| No (ref) | 309 (85.6%) | 2232 (85.2%) | | | | |
| Yes | 52 (14.4%) | 389 (14.8%) | 0.97 (0.71-1.32) | 0.83 | 0.98 (0.71-1.34) | 0.89 |
| **Head injury** | | | | | | |
| No (ref) | 328 (90.9%) | 2428 (92.6%) | | | | |
| Yes | 33 (9.1%) | 193 (7.4%) | 1.27 (0.86-1.86) | 0.23 | 1.19 (0.80-1.75) | 0.39 |
| **Migraine** | | | | | | |
| No (ref) | 286 (79.2%) | 2179 (83.1%) | | | | |
| Yes | 75 (20.8%) | 442 (16.9%) | 1.29 (0.98-1.70) | 0.07 | 1.44 (1.08-1.91) | 0.01 |
| **Meniere's disease** | | | | | | |
| No (ref) | 359 (99.4%) | 2609 (99.5%) | | | | |
| Yes | 2 (0.6%) | 12 (0.5%) | 1.21 (0.27-5.43) | 0.80 | 1.34 (0.30-6.04) | 0.70 |
| **Hyperacusis** | | | | | | |
| No (ref) | 311 (86.1%) | 2342 (89.4%) | | | | |
| Yes | 50 (13.9%) | 279 (10.6%) | 1.35 (0.98-1.86) | 0.07 | 1.40 (1.01-1.94) | 0.04 |
| **Chronic ear infection** | | | | | | |
| No (ref) | 338 (93.6%) | 2475 (94.4%) | | | | |
| Yes | 23 (6.4%) | 146 (5.6%) | 1.15 (0.73-1.82) | 0.54 | 1.19 (0.75-1.87) | 0.46 |
| **Occupational noise** | | | | | | |
| Quiet workplace (ref) | 208 (57.6%) | 1714 (65.4%) | | | | |
| Noisy workplace, always protected | 19 (5.3%) | 106 (4.0%) | 1.48 (0.89-2.46) | 0.13 | 1.30 (0.76-2.22) | 0.34 |
| Noisy workplace, frequently protected | 30 (8.3%) | 187 (7.1%) | 1.32 (0.88-1.99) | 0.18 | 1.15 (0.73-1.80) | 0.55 |
| Noisy workplace, occasionally protected | 66 (18.3%) | 323 (12.3%) | 1.66 (1.23-2.24) | <0.005 | 1.48 (1.05-2.08) | 0.02 |
| Noise workplace, never protected | 38 (10.5%) | 291 (11.1%) | 1.10 (0.77-1.59) | 0.59 | 1.03 (0.71-1.50) | 0.86 |
| **Better ear 4FA, mean (SD)** | 13.40 (9.28) | 12.32 (9.14) | 1.01 (1.00-1.02) | 0.04 | 1.01 (1.00-1.02) | 0.06 |
| Normal | 288 (79.8%) | 2163 (82.5%) | | | | |
| Mild | 64 (17.7%) | 406 (15.5%) | | | | |
| Moderate | 7 (1.9%) | 41 (1.6%) | | | | |
| Moderately severe | 2 (0.6%) | 6 (0.2%) | | | | |

*(Continued)*

| | Tinnitus at phase II (n = 361) | No Tinnitus at phase II (n = 2621) | Unadjusted analysis | | Adjusted analysis | |
|---|---|---|---|---|---|---|
| | N (%) | N (%) | OR (95% CI) | p | OR (95% CI) | p |
| Severe | 0 (0.0%) | 4 (0.2%) | | | | |
| Profound | 0 (0.0%) | 1 (0.0%) | | | | |
| **Worse ear 4FA, mean (SD)** | 18.40 (12.15) | 16.71 (12.14) | 1.01 (1.00-1.02) | 0.01 | 1.01 (1.00-1.02) | 0.02 |
| Normal | 218 (60.4%) | 1824 (69.6%) | | | | |
| Mild | 117 (32.4%) | 622 (23.7%) | | | | |
| Moderate | 17 (4.7%) | 125 (4.8%) | | | | |
| Moderately severe | 3 (0.8%) | 29 (1.1%) | | | | |
| Severe | 4 (1.4%) | 8 (0.3%) | | | | |
| Profound | 1 (0.3%) | 13 (0.5%) | | | | |
| **PHQ diagnosis** | | | | | | |
| No depression (ref) | 349 (96.7%) | 2530 (96.5%) | | | | |
| Other depression | 5 (1.4%) | 49 (1.9%) | 0.74 (0.29-1.87) | 0.52 | 0.74 (0.29-1.88) | 0.53 |
| Major depression | 7 (1.9%) | 42 (1.6%) | 1.21 (0.54-2.71) | 0.65 | 1.24 (0.55-2.80) | 0.60 |
| **PHQ score, mean (SD)** | 2.45 (3.48) | 2.41 (3.27) | 1.00 (0.97-1.04) | 0.79 | 1.01 (0.98-1.04) | 0.60 |
| Minimal | 316 (87.5%) | 2275 (86.8%) | | | | |
| Mild | 33 (9.1%) | 246 (9.4%) | | | | |
| Moderate | 5 (1.4%) | 78 (3.0%) | | | | |
| Moderately severe | 3 (0.8%) | 19 (0.7%) | | | | |
| Severe | 0 (0.0%) | 3 (0.1%) | | | | |
| **DASS Depression score, mean (SD)** | 3.39 (5.43) | 3.35 (5.40) | 1.00 (0.98-1.02) | 0.88 | 1.00 (0.98-1.02) | 0.82 |
| Normal | 329 (91.1%) | 2383 (90.9%) | | | | |
| Mild | 16 (4.4%) | 113 (4.3%) | | | | |
| Moderate | 9 (2.5%) | 80 (3.1%) | | | | |
| Severe | 5 (1.4%) | 30 (1.1%) | | | | |
| Extremely severe | 2 (0.6%) | 15 (0.6%) | | | | |
| **DASS Anxiety score, mean (SD)** | 2.40 (3.69) | 2.21 (3.61) | 1.01 (0.99-1.04) | 0.33 | 1.02 (0.99-1.05) | 0.21 |
| Normal | 338 (93.6%) | 2479 (94.6%) | | | | |
| Mild | 12 (3.3%) | 52 (2.0%) | | | | |
| Moderate | 6 (1.7%) | 43 (1.6%) | | | | |
| Severe | 3 (0.8%) | 35 (1.3%) | | | | |
| Extremely severe | 2 (0.6%) | 12 (0.5%) | | | | |
| **DASS Stress score, mean (SD)** | 6.18 (6.76) | 5.99 (6.44) | 1.00 (0.99-1.02) | 0.62 | 1.01 (0.99-1.02) | 0.52 |
| Normal | 323 (89.5%) | 2399 (91.5%) | | | | |
| Mild | 21 (5.8%) | 107 (4.1%) | | | | |
| Moderate | 10 (2.8%) | 86 (3.3%) | | | | |
| Severe | 6 (1.7%) | 27 (1.0%) | | | | |
| Extremely severe | 1 (0.3%) | 2 (0.1%) | | | | |
| **Doctor diagnosed depression** | | | | | | |
| No (ref) | 264 (73.1%) | 1976 (75.4%) | | | | |
| Yes | 97 (26.9%) | 645 (24.6%) | 1.13 (0.88-1.44) | 0.35 | 1.19 (0.92-1.53) | 0.18 |
| **Doctor diagnosed anxiety** | | | | | | |
| No (ref) | 327 (90.6%) | 2454 (93.6%) | | | | |
| Yes | 34 (9.4%) | 167 (6.4%) | 1.53 (1.04-2.25) | 0.03 | 1.63 (1.10-2.41) | 0.01 |

**Table 4. Frequencies of answers to the question 'How much does tinnitus affect your daily life and activities?' in phase I and II.**

|  | Phase I (n = 881) | Phase II (n = 1084) |
|---|---|---|
| Not at all | 512 (58.1%) | 635 (58.6%) |
| Occasionally | 293 (33.3%) | 354 (32.7%) |
| Frequently | 51 (5.8%) | 59 (5.4%) |
| Constantly | 22 (2.5%) | 36 (3.3%) |
| Missing | 3 (0.3%) | 0 (0.0%) |

**Table 5. TRQ score per tinnitus impact category for phase II.**

|  | Mean | SD | Median | IQR | Min | Max | N (%) |
|---|---|---|---|---|---|---|---|
| **Not at all** | 1.73 | 2.91 | 0 | 3 | 0 | 23 | 632 (58.6%) |
| **Occasionally** | 7.93 | 9.96 | 5 | 9 | 0 | 69 | 352 (32.6%) |
| **Frequently** | 19.46 | 15.85 | 13 | 17.5 | 2 | 68 | 59 (5.5%) |
| **Constantly** | 30.89 | 22.33 | 24 | 30 | 1 | 81 | 36 (3.3%) |
| **Total** | 5.70 | 10.47 | 2 | 6 | 0 | 81 | 1079 (100.0%) |

**Table 6. Frequencies of answers to the question 'How does tinnitus affect your daily life and activities?' in phase I and phase II.**

|  |  | Phase II |  |  |  |  |  |
|---|---|---|---|---|---|---|---|
|  |  | Not at all | Occasionally | Frequently | Constantly | Missing |  |
| **Phase I** | **Not at all** | 292 | 95 | 11 | 2 | 0 | 400 |
|  | **Occasionally** | 74 | 140 | 23 | 13 | 0 | 250 |
|  | **Frequently** | 7 | 24 | 12 | 7 | 0 | 50 |
|  | **Constantly** | 1 | 9 | 4 | 6 | 0 | 20 |
|  | **Missing** | 1 | 1 | 1 | 0 | 0 | 3 |
|  |  | 375 | 269 | 51 | 28 | 0 |  |

When analyzing factors contributing to an increase in tinnitus impact, age was identified as a significant factor with older participants having reduced odds of the impact increasing over time (OR = 0.93, 95% CI [0.90–0.96], p = <0.005). Gender, income and education did not significantly affect tinnitus impact increasing.

Within the domain of general health, an excellent subjective general health compared to a good health, showed a significantly reduced likelihood of tinnitus impact increasing over time (OR = 0.40, 95% CI [0.18–0.88], p = 0.02). Cardiovascular disease (OR = 0.68, 95% CI [0.25–1.83], p = 044), blood pressure levels (elevated: OR = 0.88, 95% CI [0.49–1.60], p = 0.68; hypertension I: OR = 0.83, 95% CI [0.50–1.36], p = 0.46; hypertension II; OR = 0.96, 95% CI [0.51–1.81], p = 0.90), and diabetes (pre-diabetes: OR = 1.18, 95% CI [0.80–1.74], p = 0.41; diabetes: OR = 1.19, 95% CI [0.58–2.43], p = 0.64) all did not significantly affect tinnitus impact increasing over time.

Hyperacusis (OR = 1.31, 95% CI [0.86–1.99], p = 0.20), occupational noise exposure (always protected: OR = 0.96, 95% CI [0.34–2.70], p = 0.94; frequently protected: OR = 1.52, 95% CI [0.80–2.88], p = 0.20; occasionally protected: OR = 1.31, 95% CI [0.73–2.36], p = 0.37; never protected: OR = 0.93, 95% CI [0.52–1.67], p = 0.82), doctor diagnosed depression (OR = 1.10, 95% CI [0.74–1.66], p = 0.60) and doctor diagnosed anxiety (OR = 1.56, 95% CI [0.82–2.93], p = 0.17), factors within the ear-related and mental health domains, also did not significantly affect tinnitus impact increasing.

**Table 7. Outcomes of the multinomial logistic regression analysis assessing factors associated with change of tinnitus impact.**

| | Same impact (ref) n=450 | Reduced impact n=119 | | | | | Increased impact n=151 | | | | |
|---|---|---|---|---|---|---|---|---|---|---|---|
| | | | Unadjusted analysis | | Adjusted analysis | | | Unadjusted analysis | | Adjusted analysis | |
| | N (%) | N (%) | OR (95% CI) | p | OR (95% CI) | p | N (%) | OR (95% CI) | p | OR (95% CI) | p |
| **Sex** | | | | | | | | | | | |
| Female (ref) | 186 (41.3%) | 42 (35.3%) | | | | | 60 (39.7%) | | | | |
| Male | 264 (58.7%) | 77 (64.7%) | 1.29 (0.85-1.97) | 0.23 | 1.29 (0.85-1.96) | 0.24 | 91 (60.3%) | 1.07 (0.73-1.56) | 0.73 | 1.10 (0.75-1.62) | 0.61 |
| Age in years, mean (SD) | 59.73 (5.41) | 59.92 (5.59) | 1.01 (0.97-1.04) | 0.74 | 1.01 (0.97-1.05) | 0.78 | 57.55 (5.71) | 0.93 (0.90-0.96) | <0.005 | 0.93 (0.90-0.96) | <0.005 |
| (45,50] | 26 (5.8%) | 7 (5.9%) | | | | | 17 (11.3%) | | | | |
| (50,55] | 68 (15.1%) | 18 (15.1%) | | | | | 36 (23.8%) | | | | |
| (55,60] | 118 (26.2%) | 26 (21.8%) | | | | | 36 (23.8%) | | | | |
| (60,65] | 160 (35.6%) | 47 (39.5%) | | | | | 47 (31.1%) | | | | |
| (65,70] | 78 (17.3%) | 21 (17.6%) | | | | | 15 (9.9%) | | | | |
| **Marital status** | | | | | | | | | | | |
| Single | 20 (4.4%) | 4 (3.4%) | 0.75 (0.25-2.24) | 0.60 | 0.77 (0.26-2.33) | 0.65 | 12 (7.9%) | 1.89 (0.90-4.00) | 0.10 | 1.72 (0.80-3.69) | 0.16 |
| Married (ref) | 344 (76.4%) | 92 (77.3%) | | | | | 109 (72.7%) | | | | |
| Widowed | 11 (2.4%) | 5 (4.2%) | 1.70 (0.58-5.01) | 0.34 | 1.90 (0.63-5.73) | 0.25 | 2 (1.3%) | 0.57 (0.13-2.63) | 0.48 | 0.71 (0.15-3.33) | 0.67 |
| Divorced | 31 (6.9%) | 9 (7.6%) | 1.09 (0.50-2.36) | 0.84 | 1.11 (0.51-2.43) | 0.79 | 9 (6.0%) | 0.92 (0.42-1.98) | 0.82 | 0.99(0.45-2.16) | 0.98 |
| Separated | 11 (2.4%) | 2 (1.7%) | 0.68 (0.15-3.12) | 0.62 | 0.71 (0.15-3.28) | 0.66 | 5 (3.3%) | 1.43 (0.49-4.22) | 0.51 | 1.29 (0.43-3.86) | 0.65 |
| De facto | 33 (7.3%) | 7 (5.9%) | 0.79 (0.34-1.85) | 0.59 | 0.80 (0.34-1.87) | 0.60 | 14 (9.3%) | 1.34 (0.69-2.59) | 0.39 | 1.29 (0.66-2.53) | 0.45 |
| **Income** | | | | | | | | | | | |
| Less than $20,000 | 24 (5.3%) | 14 (11.8%) | 2.36 (1.07-5.17) | 0.03 | 2.77 (1.21-6.31) | 0.02 | 7 (4.6%) | 0.90 (0.36-2.27) | 0.82 | 1.38 (0.53-3.62) | 0.51 |
| $20,001-$40,000 | 67 (14.9%) | 20 (16.8%) | 1.21 (0.62-2.33) | 0.58 | 1.39 (0.69-2.76) | 0.36 | 23 (15.2%) | 1.06 (0.58-1.95) | 0.85 | 1.52 (0.80-2.90) | 0.21 |
| $40,001-$60,000 | 80 (17.9%) | 19 (16.0%) | 0.96 (0.50-1.85) | 0.90 | 1.04 (0.53-2.05) | 0.90 | 28 (18.5%) | 1.08 (0.60-1.93) | 0.79 | 1.41 (0.77-2.57) | 0.26 |
| $60,001-$80,000 | 74 (16.4%) | 21 (17.6%) | 1.15 (0.60-2.19) | 0.70 | 1.23 (0.64-2.36) | 0.54 | 22 (14.6%) | 0.92 (0.50-1.70) | 0.78 | 1.09 (0.58-2.04) | 0.79 |
| $80,000-$100,000 | 58 (12.9%) | 6 (5.0%) | 0.42 (0.16-1.07) | 0.07 | 0.43 (0.17-1.12) | 0.08 | 19 (12.6%) | 1.01 (0.53-1.93) | 0.97 | 1.04 (0.54-2.02) | 0.90 |
| More than $100,000 (ref) | 105 (23.3%) | 26 (21.8%) | | | | | 34 (22.5%) | | | | |
| Unknown | 42 (9.3%) | 13 (10.9%) | 1.25 (0.59-2.66) | 0.56 | 1.38 (0.63-3.00) | 0.42 | 18 (11.9%) | 1.32 (0.67-2.60) | 0.42 | 1.86 (0.92-3.77) | 0.08 |
| **Highest education** | | | | | | | | | | | |
| Did not go to school | 0 (0.0%) | 0 (0.0%) | NA | NA | NA | NA | 0 (0.0%) | NA | NA | NA | NA |
| Primary school | 4 (0.9%) | 1 (0.8%) | 0.92 (0.10-8.34) | 0.94 | 0.91 (0.10-8.29) | 0.93 | 0 (0.0%) | NA | NA | NA | NA |
| Secondary school (ref) | 216 (48.0%) | 59 (49.6%) | | | | | 73 (48.3%) | | | | |
| University | 68 (15.1%) | 28 (23.5%) | 0.70 (0.43-1.13) | 0.15 | 1.50 (0.89-2.54) | 0.13 | 29 (19.2%) | 0.89 (0.59-1.36) | 0.60 | 1.26 (0.75-2.11) | 0.38 |
| Other educational institute | 162 (36.0%) | 31 (26.1%) | 1.51 (0.89-2.55) | 0.13 | 0.69 (0.43-1.12) | 0.14 | 49 (32.5%) | 1.26 (0.76-2.10) | 0.37 | 0.87 (0.57-1.33) | 0.52 |
| BMI, mean (SD) | 28.19 (4.75) | 29.51 (5.35) | 1.05 (1.01-1.10) | 0.01 | 1.06 (1.01-1.10) | 0.01 | 28.78 (5.12) | 1.03 (0.99-1.06) | 0.19 | 1.03 (0.99-1.07) | 0.11 |
| Underweight | 1 (0.2%) | 0 (0.0%) | | | | | 1 (0.7%) | | | | |
| Normal | 110 (24.4%) | 23 (19.3%) | | | | | 29 (19.2%) | | | | |
| Overweight | 215 (47.8%) | 50 (42.0%) | | | | | 70 (46.4%) | | | | |
| Obese | 124 (27.6%) | 46 (38.7%) | | | | | 51 (33.8%) | | | | |

*(Continued)*

| | Same impact (ref) n=450 | Reduced impact n=119 | | | | | Increased impact n=151 | | | | |
|---|---|---|---|---|---|---|---|---|---|---|---|
| | | | Unadjusted analysis | | Adjusted analysis | | | Unadjusted analysis | | Adjusted analysis | |
| | N (%) | N (%) | OR (95% CI) | p | OR (95% CI) | p | N (%) | OR (95% CI) | p | OR (95% CI) | p |
| **Waist in cm, mean (SD)** | 94.13 (12.90) | 96.80 (14.52) | 1.02 (1.00-1.03) | 0.05 | 1.01 (1.00-1.03) | 0.10 | 94.71 (13.00) | 1.00 (0.99-1.02) | 0.64 | 1.01 (0.99-1.02) | 0.32 |
| **General health** | | | | | | | | | | | |
| Poor | 3 (0.7%) | 1 (0.8%) | 0.99 (0.10-9.74) | 0.99 | 1.08 (0.11-10.60) | 0.95 | 2 (1.3%) | 1.84 (0.30-11.24) | 0.51 | 2.01 (0.32-12.57) | 0.46 |
| Fair | 34 (7.6%) | 15 (12.6%) | 1.32 (0.67-2.58) | 0.42 | 1.34 (0.68-2.63) | 0.40 | 16 (10.6%) | 1.30 (0.67-2.50) | 0.44 | 1.34 (0.69-2.60) | 0.40 |
| Good (ref) | 179 (39.8%) | 60 (50.4%) | | | | | 65 (43.0%) | | | | |
| Very good | 175 (38.9%) | 37 (31.1%) | 0.63 (0.40-1.00) | 0.049 | 0.63 (0.40-1.01) | 0.049 | 60 (39.7%) | 0.94 (0.63-1.42) | 0.78 | 0.94 (0.62-1.42) | 0.77 |
| Excellent | 59 (13.1%) | 6 (5.0%) | 0.30 (0.12-0.74) | 0.01 | 0.30 (0.12-0.73) | 0.01 | 8 (5.3%) | 0.37 (0.17-0.82) | 0.01 | 0.40 (0.18-0.88) | 0.02 |
| **Smoking in pack years, mean (SD)** | 7.09 (11.81) | 10.69 (15.67) | 1.02 (1.01-1.03) | 0.01 | 1.02 (1.00-1.03) | 0.01 | 10.18 (15.48) | 1.52 (1.00-2.33) | 0.01 | 1.02 (1.01-1.04) | <0.005 |
| Never smokers | 228 (50.7%) | 48 (40.3%) | | | | | 63 (41.7%) | | | | |
| Light smokers | 163 (36.2%) | 51 (42.9%) | | | | | 64 (42.4%) | | | | |
| Moderate smokers | 46 (10.2%) | 12 (10.1%) | | | | | 13 (8.6%) | | | | |
| Heavy smokers | 13 (2.9%) | 8 (6.7%) | | | | | 11 (7.3%) | | | | |
| **CVD** | | | | | | | | | | | |
| No (ref) | 427 (94.9%) | 100 (84.0%) | | | | | 146 (96.7%) | | | | |
| Yes | 23 (5.1%) | 19 (16.0%) | 3.53 (1.85-8.73) | <0.005 | 3.41 (1.78-6.54) | <0.005 | 5 (3.3%) | 0.64 (0.24-1.70) | 0.37 | 0.68 (0.25-1.83) | 0.44 |
| **Blood pressure** | | | | | | | | | | | |
| Normal (ref) | 85 (18.9%) | 30 (25.2%) | | | | | 35 (23.2%) | | | | |
| Elevated | 83 (18.4%) | 23 (19.3%) | 0.84 (0.45-1.56) | 0.58 | 0.74 (0.39-1.38) | 0.34 | 28 (18.5%) | 0.79 (0.44-1.41) | 0.42 | 0.88 (0.49-1.60) | 0.68 |
| Hypertension I | 212 (47.1%) | 50 (42.0%) | 0.66 (0.40-1.11) | 0.12 | 0.61 (0.36-1.05) | 0.07 | 65 (43.0%) | 0.72 (0.44-1.16) | 0.17 | 0.83 (0.50-1.36) | 0.46 |
| Hypertension II | 70 (15.6%) | 16 (13.4%) | 0.61 (0.30-1.22) | 0.16 | 0.60 (0.30-1.21) | 0.15 | 23 (15.2%) | 0.78 (0.42-1.43) | 0.42 | 0.96 (0.51-1.81) | 0.90 |
| **Diabetes** | | | | | | | | | | | |
| No (ref) | 223 (49.6%) | 47 (39.5%) | | | | | 69 (45.7%) | | | | |
| Pre-DM | 192 (42.7%) | 55 (46.2%) | 1.36 (0.88-2.10) | 0.17 | 1.35 (0.87-2.08) | 0.18 | 70 (46.4%) | 1.18 (0.80-1.73) | 0.40 | 1.18 (0.80-1.74) | 0.41 |
| DM | 35 (7.8%) | 17 (14.3%) | 2.30 (1.19-4.46) | 0.01 | 2.25 (1.16-4.36) | 0.02 | 12 (7.9%) | 1.11 (0.55-2.25) | 0.78 | 1.19 (0.58-2.43) | 0.64 |
| **Cancer** | | | | | | | | | | | |
| No (ref) | 381 (84.7%) | 90 (75.6%) | | | | | 129 (85.4%) | | | | |
| Yes | 69 (15.3%) | 29 (24.4%) | 1.78 (1.09-2.90) | 0.02 | 1.78 (1.08-2.93) | 0.02 | 22 (14.6%) | 0.94 (0.56-1.58) | 0.82 | 1.09 (0.64-1.84) | 0.76 |
| **Head injury** | | | | | | | | | | | |
| No (ref) | 411 (91.3%) | 109 (91.6%) | | | | | 141 (93.4%) | | | | |
| Yes | 39 (8.7%) | 10 (8.4%) | 0.97 (0.47-2.00) | 0.93 | 0.93 (0.45-1.93) | 0.85 | 10 (6.6%) | 0.75 (0.36-1.54) | 0.43 | 0.71 (0.34-1.48) | 0.36 |
| **Migraine** | | | | | | | | | | | |
| No (ref) | 371 (82.4%) | 102 (85.7%) | | | | | 127 (84.1%) | | | | |
| Yes | 79 (17.6%) | 17 (14.3%) | 0.89 (0.52-1.54) | 0.68 | 0.96 (0.55-1.68) | 0.89 | 24 (15.9%) | 0.84 (0.51-1.40) | 0.51 | 0.77 (0.46-1.31) | 0.34 |
| **Meniere's disease** | | | | | | | | | | | |
| No (ref) | 438 (97.3%) | 115 (96.6%) | | | | | 148 (98.0%) | | | | |
| Yes | 12 (2.7%) | 4 (3.4%) | 1.27 (0.40-4.01) | 0.68 | 1.31 (0.41-4.14) | 0.65 | 3 (2.0%) | 0.73 (0.21-2.66) | 0.64 | 0.81 (0.22-2.94) | 0.75 |

(Continued)

Table 7. (Continued)

| | Same impact (ref) n=450 | Reduced impact n=119 | | | | | Increased impact n=151 | | | | |
|---|---|---|---|---|---|---|---|---|---|---|---|
| | N (%) | N (%) | Unadjusted analysis OR (95% CI) | p | Adjusted analysis OR (95% CI) | p | N (%) | Unadjusted analysis OR (95% CI) | p | Adjusted analysis OR (95% CI) | p |
| **Hyperacusis** | | | | | | | | | | | |
| No (ref) | 338 (75.1%) | 82 (68.9%) | | | | | 107 (70.9%) | | | | |
| Yes | 112 (24.9%) | 37 (31.1%) | 1.36 (0.87-2.12) | 0.17 | 1.38 (0.88-2.15) | 0.16 | 44 (29.1%) | 1.24 (0.82-1.88) | 0.30 | 1.31 (0.86-1.99) | 0.20 |
| **Chronic ear infection** | | | | | | | | | | | |
| No (ref) | 413 (91.8%) | 105 (88.2%) | | | | | 141 (93.4%) | | | | |
| Yes | 37 (8.2%) | 14 (11.8%) | 1.49 (0.78-2.85) | 0.23 | 1.52 (0.79-2.93) | 0.21 | 10 (6.6%) | 0.79 (0.39-1.63) | 0.53 | 0.74 (0.36-1.55) | 0.43 |
| **Occupational noise** | | | | | | | | | | | |
| Quiet workplace (ref) | 193 (42.9%) | 37 (31.1%) | | | | | 58 (38.4%) | | | | |
| Noisy work, always protected | 20 (4.4%) | 8 (6.7%) | 2.08 (0.85-5.07) | 0.11 | 2.12 (0.81-5.55) | 0.13 | 6 (4.0%) | 0.99 (0.38-2.59) | 0.99 | 0.96 (0.34-2.70) | 0.94 |
| Noisy work, frequently protected | 64 (14.2%) | 14 (11.8%) | 1.12 (0.57-2.20) | 0.75 | 1.15 (0.53-2.50) | 0.72 | 30 (19.9%) | 1.53 (0.91-2.58) | 0.11 | 1.52 (0.80-2.88) | 0.20 |
| Noisy work, occasionally protected | 96 (21.3%) | 35 (29.4%) | 1.89 (1.12-3.19) | 0.02 | 1.91 (1.02-3.57) | 0.049 | 36 (23.8%) | 1.24 (0.77-2.01) | 0.38 | 1.31 (0.73-2.36) | 0.37 |
| Noisy work, never protected | 77 (17.1%) | 25 (11.8%) | 1.68 (0.95-2.99) | 0.07 | 1.69 (0.94-3.04) | 0.08 | 21 (13.9%) | 0.90 (0.51-1.59) | 0.72 | 0.93 (0.52-1.67) | 0.82 |
| **Better ear 4FA, mean (SD)** | 17.56 (9.92) | 20.88 (12.02) | 1.03 (1.01-1.05) | <0.005 | 1.03 (1.01-1.05) | <0.005 | 17.53 (11.02) | 1.00 (0.98-1.02) | 0.98 | 1.02 (1.00-1.04) | 0.10 |
| Normal | 275 (61.1%) | 56 (47.1%) | | | | | 93 (61.6%) | | | | |
| Mild | 150 (33.3%) | 51 (42.9%) | | | | | 48 (31.8%) | | | | |
| Moderate | 24 (5.3%) | 10 (8.4%) | | | | | 7 (4.6%) | | | | |
| Moderately severe | 1 (0.2%) | 2 (1.7%) | | | | | 3 (2.0%) | | | | |
| Severe | 0 (0.0%) | 0 (0.0%) | | | | | 0 (0.0%) | | | | |
| Profound | 0 (0.0%) | 0 (0.0%) | | | | | 0 (0.0%) | | | | |
| **Worse ear 4FA, mean (SD)** | 23.43 (13.85) | 28.66 (17.04) | 1.02 (1.01-1.04) | <0.005 | 1.02 (1.01-1.04) | <0.005 | 28.53 (14.77) | 1.00 (0.99-1.02) | 0.65 | 1.01 (1.00-1.03) | 0.049 |
| Normal | 197 (43.8%) | 41 (34.5%) | | | | | 65 (43.0%) | | | | |
| Mild | 185 (41.1%) | 45 (37.8%) | | | | | 57 (37.7%) | | | | |
| Moderate | 48 (10.7%) | 22 (18.5%) | | | | | 20 (13.2%) | | | | |
| Moderately severe | 12 (2.7%) | 7 (5.9%) | | | | | 7 (4.6%) | | | | |
| Severe | 3 (0.7%) | 2 (1.7%) | | | | | 1 (0.7%) | | | | |
| Profound | 5 (1.1%) | 2 (1.7%) | | | | | 1 (0.7%) | | | | |
| **PHQ diagnosis** | | | | | | | | | | | |
| No depression (ref) | 423 (94.0%) | 117 (98.3%) | | | | | 142 (94.0%) | | | | |
| Other depression | 12 (2.7%) | 1 (0.8%) | 0.30 (0.04-2.34) | 0.25 | 0.30 (0.04-2.35) | 0.25 | 4 (2.6%) | 0.99 (0.32-3.13) | 0.99 | 0.82 (0.25-2.62) | 0.73 |
| Major depression | 15 (3.3%) | 1 (0.8%) | 0.24 (0.03-1.84) | 0.17 | 0.25 (0.03-1.92) | 0.18 | 5 (3.3%) | 0.99 (0.35-2.78) | 0.99 | 0.98 (0.35-2.77) | 0.97 |

*(Continued)*

**Table 7.** (Continued)

| | Same impact (ref) n=450 | Reduced impact n=119 | | | | | Increased impact n=151 | | | | | |
|---|---|---|---|---|---|---|---|---|---|---|---|---|
| | | | Unadjusted analysis | | Adjusted analysis | | | Unadjusted analysis | | Adjusted analysis | | |
| | N (%) | N (%) | OR (95% CI) | p | OR (95% CI) | p | N (%) | OR (95% CI) | p | OR (95% CI) | p | |
| **PHQ score, mean (SD)** | 2.86 (3.97) | 3.34 (3.90) | 1.03 (0.98-1.08) | 0.22 | 1.04 (0.99-1.09) | 0.15 | 3.46 (3.80) | 1.04 (0.99-1.09) | 0.22 | 1.03 (0.98-1.08) | 0.21 | |
| Minimal | 378 (84.0%) | 94 (79.0%) | | | | | 116 (76.8%) | | | | | |
| Mild | 47 (10.4%) | 17 (14.3%) | | | | | 27 (17.9%) | | | | | |
| Moderate | 14 (3.1%) | 7 (5.9%) | | | | | 3 (2.0%) | | | | | |
| Moderately severe | 9 (2.0%) | 1 (0.8%) | | | | | 5 (3.3%) | | | | | |
| Severe | 2 (0.4%) | 0 (0.0%) | | | | | 0 (0.0%) | | | | | |
| **DASS depression score, mean (SD)** | 4.30 (6.44) | 4.32 (5.37) | 1.00 (0.97-1.03) | 0.97 | 1.00 (0.97-1.03) | 0.93 | 5.36 (7.21) | 1.02 (1.00-1.05) | 0.08 | 1.02 (0.99-1.05) | 0.16 | |
| Normal | 395 (87.8%) | 104 (87.4%) | | | | | 125 (82.8%) | | | | | |
| Mild | 28 (6.2%) | 8 (6.7%) | | | | | 14 (9.3%) | | | | | |
| Moderate | 10 (2.2%) | 5 (4.2%) | | | | | 6 (4.0%) | | | | | |
| Severe | 12 (2.7%) | 2 (1.7%) | | | | | 3 (2.0%) | | | | | |
| Extremely severe | 5 (1.1%) | 0 (0.0%) | | | | | 3 (2.0%) | | | | | |
| **DASS anxiety score, mean (SD)** | 2.76 (4.09) | 3.24 (4.07) | 1.03 (0.98-1.08) | 0.24 | 1.03 (0.99-1.08) | 0.18 | 3.18 (4.15) | 1.03 (0.98-1.07) | 0.26 | 1.02 (0.98-1.07) | 0.29 | |
| Normal | 415 (92.2%) | 106 (89.1%) | | | | | 135 (89.4%) | | | | | |
| Mild | 9 (2.0%) | 9 (7.6%) | | | | | 5 (3.3%) | | | | | |
| Moderate | 15 (3.3%) | 1 (0.8%) | | | | | 7 (4.6%) | | | | | |
| Severe | 10 (2.2%) | 2 (1.7%) | | | | | 3 (2.0%) | | | | | |
| Extremely severe | 1 (0.2%) | 1 (0.8%) | | | | | 1 (0.7%) | | | | | |
| **DASS stress score, mean (SD)** | 6.64 (7.06) | 7.31 (6.80) | 1.01 (0.99-1.04) | 0.34 | 1.01 (1.00-1.05) | 0.32 | 7.97 (7.25) | 1.03 (1.00-1.05) | 0.04 | 1.02 (1.00-1.05) | 0.08 | |
| Normal | 402 (89.3%) | 106 (89.1%) | | | | | 127 (84.1%) | | | | | |
| Mild | 24 (5.3%) | 4 (3.4%) | | | | | 11 (7.3%) | | | | | |
| Moderate | 16 (3.6%) | 8 (6.7%) | | | | | 11 (7.3%) | | | | | |
| Severe | 4 (0.9%) | 1 (0.8%) | | | | | 1 (0.7%) | | | | | |
| Extremely severe | 4 (0.9%) | 0 (0.0%) | | | | | 1 (0.7%) | | | | | |
| **Doctor diagnosed depression** | | | | | | | | | | | | |
| No (ref) | 313 (69.9%) | 77 (64.7%) | | | | | 101 (66.9%) | | | | | |
| Yes | 137 (30.4%) | 42 (35.3%) | 1.25 (0.81-1.91) | 0.31 | 1.29 (0.84-1.99) | 0.24 | 50 (33.1%) | 1.13 (0.76-1.68) | 0.54 | 1.11 (0.74-1.66) | 0.60 | |
| **Doctor diagnosed anxiety** | | | | | | | | | | | | |
| No (ref) | 416 (92.4%) | 105 (88.2%) | | | | | 135 (89.4%) | | | | | |
| Yes | 34 (7.6%) | 14 (11.8%) | 1.63 (0.84-3.15) | 0.15 | 1.66 (0.86-3.21) | 0.13 | 16 (10.6%) | 1.45 (0.78-2.71) | 0.24 | 1.56 (0.82-2.93) | 0.17 | |

## Discussion

By analyzing longitudinal data obtained from the general population, this study aimed to (1) assess the incidence of tinnitus and identify risk factors for the development of tinnitus, and to (2) assess the impact of tinnitus and change in impact over time and identify factors associated with this change.

In this cohort of 3863 Western Australian baby boomers born between 1946 and 1964, the prevalence of tinnitus increased from 22.8% during the initial phase to 28.1% at the follow-up six years later. The incidence of new-onset tinnitus over this period was 12.11%, corresponding to an incidence of 2017.66 per 100,000 person-years. This incidence exceeds the pooled incidence of 1164 per 100,000 person-years reported in a recent meta-analysis based on six studies among adults in Western populations [3]. This difference may be attributed to heterogeneity in tinnitus definitions and study populations. Other studies with similar age demographics show incidence rates more in line with the current study: the UK Biobank reported an 8.7% incidence over 4 years [29], the Blue Mountains study reported 18% over 5 years [16], and the Epidemiology of Hearing Loss study reported 12.7% over 10 years [10].

### Risk factors for the development of tinnitus

The examination of risk factors for developing tinnitus identified demographic, general health, ear-related health and mental health factors that were significantly associated with an increased odd of developing tinnitus. Gender was identified as a significant demographic risk factor, with males showing greater odds of developing tinnitus. This result is consistent with the results of Nondahl et al., (2010) from the Epidemiology of Hearing Loss study where male sex was also found to be associated with an increased risk of incident tinnitus [10]. This may be explained by factors such as higher exposure to occupational noise and other health-related factors that tend to be more pronounced in males. However, other studies on risk factor analysis for tinnitus incidence have not consistently demonstrated a significant difference between males and females [6,16]. Age was not identified as a significant factor for tinnitus incidence in our study. This may seem unexpected, as previous research generally reports increasing tinnitus prevalence with age [3,4]. However, our study specifically focused on a middle-aged population (46–64 years) to examine risk factors within this age group. Within this relatively narrow range, age-related differences in tinnitus risk may be less pronounced than in studies including both younger and older adults, which could explain the lack of association in our cohort.

Among the variables in the general health domain, a higher BMI and larger waist circumference were associated with increased odds of tinnitus development. While these obesity-related factors have also been found to be associated with tinnitus in cross-sectional studies [5], our results contrast with other analytical observational studies showing a lower risk in obese participants [10,30]. This discrepancy may reflect differences in population characteristics across studies, but may also be confounded by other lifestyle factors that influence both body composition and tinnitus. Moreover, the small odds ratios observed for both factors highlight the importance of considering multiple interacting variables when examining this link and interpreting its clinical significance. A history of cardiovascular disease, as well as objective measures of hypertension and diabetes, were not identified as risk factors for the development of tinnitus in our study. This finding is particularly noteworthy given that cardiometabolic factors are often included in tinnitus studies and have been found to be associated with tinnitus in multiple cross-sectional studies [5]. However, the meta-analysis by Biswas et al. (2023), which excluded those cross-sectional studies and only focused on analytical observational studies, found no evidence for an association between tinnitus and heart failure, stroke, or hypertension, while diabetes was negatively associated with tinnitus [6]. These results underscore the complex and potentially time-dependent relationship between tinnitus and cardiometabolic factors. Migraine was the only other general health factor significantly associated with the development of tinnitus in this study, while smoking, cancer history, and head injury showed no significant association.

Within the domain of ear-related health, a higher hearing threshold in the worse ear, hyperacusis, and exposure to noisy work environments while only occasionally protected emerged as significant risk factors. These findings support the established link between hearing loss and tinnitus [5–9], and underscore the importance of hearing protection in

occupational settings. Of note is that exposure to noisy work environments without wearing hearing protection did not emerge as a significant risk factor. This may further underline the multifactorial nature of tinnitus.

Within the domain of mental health, our study found that clinician-diagnosed anxiety is a significant risk factor for the development of tinnitus, while clinician-diagnosed depression and both self-reported measures of anxiety and depression scales, did not show a significant association. This result is somewhat surprising, as depression has been extensively described as a risk factor for tinnitus in previous research [5,6]. The discrepancy could be attributed to differences in study design, population characteristics, or the complex temporal relationship between depression and tinnitus onset. Overall, these results emphasize the complex interplay between tinnitus and mental health, warranting further research to explore these relationships more deeply.

## Tinnitus impact and change in impact

In this study we were also able to analyze the perceived impact of tinnitus and change in impact over time. The perceived impact of tinnitus varied among participants. At baseline, 41.6% reported that tinnitus affected their daily life and activities at least occasionally, with a similar percentage of 41.4% during follow-up. The proportion of participants reporting that tinnitus frequently or constantly impacted them, was 8.3% at baseline and 8.7% at follow-up. This suggests that while tinnitus can be disruptive for some, the majority of people do not experience severe interference with daily functioning.

When examining changes in impact over time for those who experienced tinnitus during both phases, 62.2% reported no change in impact on their daily life and activities, while 21.4% reported an increased impact and 16.5% noted a decrease. This pattern is consistent with findings from other longitudinal studies. For instance, Gopinath et al. (2010) reported that 60% of their cohort experienced no change in tinnitus impact over time, while 21.2% experienced an increase and 18.8% a decrease in tinnitus impact [16]. Similarly, Dawes et al. (2010) reported that 81.8% of their participants experienced no change, with 9% noting an increase and 9% a decrease in impact [29]. This consistent trend across studies suggest that while tinnitus impact remains stable for the majority of individuals over time, a minority experiences either worsening or improvement of symptoms.

Notably, 21.9% of participants who experienced tinnitus at baseline no longer reported the presence of tinnitus at the follow-up measurement six years later. Similarly, Dawes et al. (2010) found that 18% of the participants in their population no longer reported tinnitus after the study period of 4 years [29] and Gopinath et al. (2010) reported 16.2% over 5 years [16]. Of those no longer reporting tinnitus during follow-up in our study, the majority did not experience any impact (70.9%) or only occasional impact (27.2%) at baseline. It is important to note that our study did not examine the effects of specific treatments on tinnitus impact. Therefore, the observed changes could reflect the natural course of tinnitus over time but also effect of interventions.

## Factors associated with change in tinnitus impact

Our analysis of factors contributing to these changes in tinnitus impact over time revealed several statistically significant associations. Higher hearing thresholds at the worse ear and smoking were significantly associated with greater odds of the impact of tinnitus increasing over time. Conversely, advanced age and self-reported 'excellent' general health were associated with lower odds of increased impact. These findings suggest that both audiological and lifestyle factors may play a role in the progression of tinnitus severity.

Factors associated with a decrease in tinnitus impact over time presented a more complex picture. A low income, higher BMI, a history of cancer, cardiovascular disease, and diabetes were all associated with increased odds of improvement over time, as was exposure to noisy work environments while only occasionally protected and higher hearing thresholds in both the worse and better ear. Paradoxically, both smoking and self-reported excellent general health were also associated with decreased odds of improvement.

The limited number of longitudinal studies examining factors influencing changes in tinnitus impact over time constrains our ability to contextualize these findings fully. To our knowledge, only Dawes et al. (2010) explored this topic, identifying associations between increased bothersomeness of tinnitus and being female, not drinking alcohol, and having higher better ear speech recognition thresholds [29]. Our results both complement and expand upon these findings, highlighting the need for further research in this area to better understand the various factors that influence tinnitus progression and to create more targeted and effective interventions.

### Implications, limitations and future research

The findings of this study provide important insights into the potential for preventive strategies in tinnitus management. As tinnitus is a multifaceted problem, prevention should not just target one factor, but rather consider and address multiple factors simultaneously. Effective prevention strategies should encompass hearing protection to mitigate exposure to harmful noise levels, promote lifestyle modifications that support overall health, and provide mental health support to address psychological factors associated with tinnitus. By adopting a comprehensive approach to prevention, it may be possible to reduce the incidence of tinnitus and improve the overall well-being of individuals at risk.

Some methodological limitations of this study should, however, be noted. While the study benefits from a large sample derived from the general population and employs a longitudinal design, the analyses of risk factors for tinnitus presence and change in impact involve smaller subsets of participants. This is due to the division of the total sample into those with and without tinnitus, and further separation in groups based on different characteristics. This reduces the ability to detect statistically significant associations due to a lack of statistical power. Furthermore, the assessment of tinnitus impact relied on a single-item question in both phases, supplemented by the multi-item TRQ in the follow-up phase. Since no multi-item questionnaire was available at the start of the study, changes in impact were based on the single-item question with categorized response options. Consequently, the results of changes in impact may differ if a tinnitus impact questionnaire with a continuous scale score, supported by cut-off scores based on clinically important change, was used. However, the data from the TRQ during the follow-up suggests that the single-item question may be sufficient, as the patterns observed in TRQ scores align closely with the self-reported impact categories. This is consistent with a previous study showing that a single-item question was associated with the impact scales defined for the Tinnitus Functional Index [31]. Thus, in population-based studies, where questionnaire length may be a limiting factor, a single-item question could be a sufficient and practical alternative.

Future research should incorporate measures of both tinnitus presence and impact in longitudinal studies across diverse populations and age ranges. This approach would provide valuable insights in tinnitus development and its progression over time, and could guide the design of targeted preventative and interventional strategies tailored to different demographic groups.

### Conclusion

The high incidence of tinnitus and its notable impact on daily life emphasize the importance of gaining a deeper understanding of the broad range of identified risk factors for developing tinnitus and change in tinnitus impact. The multifaceted nature of these factors, encompassing hearing-related issues, general health conditions and psychological aspects, underscore the complexity of tinnitus etiology and impact. By gaining more insight into these factors, we can inform future research efforts aimed at developing targeted prevention and intervention strategies tailored to specific demographic groups.

### Author contributions

**Conceptualization:** Denise Fuchten, Inge Stegeman, Robert H. Eikelboom, Adriana L. Smit.

**Data curation:** Robert H. Eikelboom, Michael L. Hunter.

Formal analysis: Denise Fuchten, Yinan Mao.

Methodology: Denise Fuchten, Inge Stegeman, Yinan Mao, Robert H. Eikelboom, Adriana L. Smit.

Project administration: Denise Fuchten, Inge Stegeman, Robert H. Eikelboom, Michael L. Hunter, Adriana L. Smit.

Resources: Robert H. Eikelboom, Michael L. Hunter.

Supervision: Inge Stegeman, Adriana L. Smit.

Writing – original draft: Denise Fuchten.

Writing – review & editing: Denise Fuchten, Inge Stegeman, Yinan Mao, Robert H. Eikelboom, Michael L. Hunter, Adriana L. Smit.

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
