## [Decision Letter · Decision Letter 0]

24 Oct 2025

PONE-D-25-16358
Longitudinal analysis of the risk factors for onset and change in tinnitus in the Busselton Healthy Ageing Study
PLOS ONE

Dear Dr. Fuchten,

Thank you for submitting your manuscript to PLOS ONE. After careful consideration, we feel that it has merit but does not fully meet PLOS ONE’s publication criteria as it currently stands. Therefore, we invite you to submit a revised version of the manuscript that addresses the points raised during the review process.

**ACADEMIC EDITOR: Please see reviewer's comments and submit your responses with the revised manuscript**

We look forward to receiving your revised manuscript.

Kind regards,

Gauri Mankekar, MD,PhD,FACS

Academic Editor

PLOS ONE

Journal Requirements:

4. In the online submission form, you indicated that your data is available only on request from a third party. Please note that your Data Availability Statement is currently missing [the name of the third party contact or institution / contact details for the third party, such as an email address or a link to where data requests can be made]. Please update your statement with the missing information.

“Core funding for data collection: Grants from the Government of Western Australia (Department of Jobs, Tourism, Science and Innovation), the Commonwealth Government (Department of Health), the City of Busselton and from private donations to the Busselton Population Medical Research Institute. *In-kind support*: The Western Australian Country Health Service, Pathwest, Abbott, Stallergenes, Resmed, Wesnes Cognition LTD and BD Biosciences. All funding bodies had no role in the design of the study and collection, analysis, and interpretation of data and in writing the manuscript.”

Reviewers' comments:

Reviewer's Responses to Questions

**Comments to the Author**

1. Is the manuscript technically sound, and do the data support the conclusions?

Reviewer #1: Yes

Reviewer #2: Yes

2. Has the statistical analysis been performed appropriately and rigorously?

Reviewer #1: Yes

Reviewer #2: Yes

3. Have the authors made all data underlying the findings in their manuscript fully available?

Reviewer #1: Yes

Reviewer #2: Yes

4. Is the manuscript presented in an intelligible fashion and written in standard English?

Reviewer #1: Yes

Reviewer #2: Yes

5. Review Comments to the Author

Reviewer #1: Dear authors,

Congratulations with this very relevant and important manuscript. It are very interesting insights on the incidence of and risk factors for tinnitus. I only have a few minor comments as this work is scientifically and statistically sound.

comments:

- was there any correction used for multiple analyses to lower the chance of Type-I errors?

- the manuscript could benefit from insights on the use of tinnitus treatment between phase 1 and phase 2 to better understand the potential changes in tinnitus impact. I would also suggest to move lines 442-445 after line 451 as this information is also relevant to those we have no reported tinnitus at phase 2 anymore.

- some findings (e.g. lines 259-260, lines 334-335, line 395, lines 464-465) seem unexpected and I would like a bit more discussion on it with potential reasons for these unexpected findings.

- Line 390: 'incident tinnitus' -> 'tinnitus incidence'

- I suggest to elaborate on the limitations section a little further by also highlighting that not only the tinnitus impact relied on a single-item question but also the diagnosis. A validated diagnostic questionnaire would be preferred. Also, the age-range (46-64 yrs at time of phase 1) is small and a larger range could contribute to the insights.

Reviewer #2: This manuscript is technically sound, the analyses are appropriate and well conducted, and the data support the conclusions. A graphical representation / plots for major risk factors would however improve visual interpretation for readers. Although statistical associations have been well demonstrated, a short discussion on clinical significance and the impact of this study on patient counselling would strengthen the relevance of the study.

6. PLOS authors have the option to publish the peer review history of their article (what does this mean?). If published, this will include your full peer review and any attached files.

Reviewer #1: **Yes: **Dr. Remo Arts

Reviewer #2: No

---

## [Author Response · Author response to Decision Letter 1]

19 Dec 2025

19-12-2025

Dear dr. Mankekar, reviewer #1 and reviewer #2,

We thank you for your careful consideration of our manuscript and for providing constructive feedback. We have revised our manuscript to address the points raised.

Comments from the academic editor

Regarding the comments from the Academic Editor, we have now revised our manuscript to meet PLOS ONE’s style requirements, including file naming (comment 1) and have also included a complete copy of PLOS’ questionnaire on inclusivity in global research to our submission. (comment 2). Following the editor’s guidance (comment 3 and 4) we have updated our data availability statement. The revised statement reads as follows: “The data that support the findings of this study are not publicly available. However, access to the data for research can be obtained by contacting the Busselton Population Medical Research Institute busseltonhealthstudy@health.wa.gov.au.”

Our updated funding statement (comment 5) reads as follows: “Core funding for data collection: Grants from the Government of Western Australia (Department of Jobs, Tourism, Science and Innovation), the Commonwealth Government (Department of Health), the City of Busselton and from private donations to the Busselton Population Medical Research Institute. In-kind support: The Western Australian Country Health Service, Pathwest, Abbott, Stallergenes, Resmed, Wesnes Cognition LTD and BD Biosciences. All funding bodies had no role in the design of the study and collection, analysis, and interpretation of data and in writing the manuscript. No dedicated funding was obtained by the authors for the analyses conducted in the present study.”

We have removed our funding statement from the manuscript and only included the updated statement in this letter, so that it can be changed in the online submission.

Comment 6 is not applicable, as the reviewers did not suggest citing additional published work. We have verified that our reference list is complete and accurate (comment 7).

Comments from reviewer #1:

We would like to thank reviewer #1 for their feedback on our manuscript. Below, we respond to each of the reviewer’s comments in a point-by-point manner.

- Was there any correction used for multiple analyses to lower the chance of Type-I errors?

We did not apply a correction for multiple analyses. The use of such corrections is subject to ongoing methodological debate (Feise, 2002; Hooper, 2025; Rothman, 1990), as they can place disproportionate emphasis on p-values, while current recommendations increasingly encourage interpretation based on effect sizes, confidence intervals, and the overall pattern of findings. In our study, the variables included in the analyses were selected in advance based on previous literature on tinnitus risk factors, rather than by exploring a large number of available cohort variables. Given this targeted approach, we chose not to apply multiple-comparison corrections. We nevertheless interpret our findings cautiously and in the context of the broader evidence.

- The manuscript could benefit from insights on the use of tinnitus treatment between phase 1 and phase 2 to better understand the potential changes in tinnitus impact. I would also suggest to move lines 442-445 after line 451 as this information is also relevant to those we have no reported tinnitus at phase 2 anymore.

We agree that information on tinnitus treatment between phase 1 and phase 2 would provide valuable insights into potential changes in tinnitus impact. However, in this study we relied on data from the Busselton cohort, for which the dataset was pre-existing and did not include information on therapy. Although participants were asked in the phase 2 questionnaire whether they had sought help or advice for tinnitus, the timing of this help-seeking was not specified and may have occurred prior to phase 1, preventing us from drawing conclusions about treatment received between the two phases.

We also agree that lines 442–445 (due to changes in font now lines 470-473) are relevant for participants who no longer reported tinnitus in phase 2. We have therefore moved these lines to follow line 451 (now line 479) as suggested.

- Some findings (e.g. lines 259-260, lines 334-335, line 395, lines 464-465) seem unexpected and I would like a bit more discussion on it with potential reasons for these unexpected findings.

We agree that some of these findings seem unexpected. We have elaborated on age not being a significant risk factor on lines 415-421 in the discussion. Regarding the other points, these relate to factors influencing changes in tinnitus impact over time, namely: occupational noise exposure, ear-related and mental health factors, and paradoxical associations such as smoking and self-reported excellent general health. At present, we are unable to provide clear explanations for these observations. The limited number of longitudinal studies examining factors influencing changes in tinnitus impact constrains our ability to contextualize these findings fully. These observations are described and discussed in the manuscript (lines 503-510) and they highlight the need for further research to better understand factors influencing changes in tinnitus impact over time.

- Line 390: 'incident tinnitus' -> 'tinnitus incidence'

We have changed this in the manuscript (line 414).

- I suggest to elaborate on the limitations section a little further by also highlighting that not only the tinnitus impact relied on a single-item question but also the diagnosis. A validated diagnostic questionnaire would be preferred. Also, the age-range (46-64 yrs at time of phase 1) is small and a larger range could contribute to the insights.

We appreciate the reviewer’s suggestion regarding the limitations section. In our study, tinnitus was defined using the following question: “Do you experience tinnitus (sound in your ears or head) for longer than 5 minutes, which does not have an obvious cause?” This question captures the presence of tinnitus and is in line with current consensus on defining tinnitus (de Ridder et al., 2021). Multi-item questionnaires, as suggested, are only used to assess the impact or level of distress. For future research, it would be interesting to see if we could add these types of questionnaires to population studies to examine the impact of tinnitus in more detail.

We do not consider the age range a limitation, but rather as central to the study’s focus. Our objective was to examine factors related to tinnitus while considering the role of ageing. Using longitudinal data from the Busselton Healthy Ageing Study, which includes community-dwelling participants aged 46–64 at phase 1, allowed us to specifically address these research questions within this targeted age group. We did however elaborate on the findings regarding age in the context of tinnitus incidence in lines 415-421.

Comments from reviewer #2

We would also like to thank reviewer #2 for their feedback on our manuscript. Below, we respond to each of the reviewer’s comments in a point-by-point manner.

- This manuscript is technically sound, the analyses are appropriate and well conducted, and the data support the conclusions. A graphical representation / plots for major risk factors would however improve visual interpretation for readers.

We thank the reviewer for this suggestion. All results for the risk factors are presented in the tables, which include odds ratios, confidence intervals, and p-values. While graphical representations can sometimes aid visual interpretation, numeric data in tables allow readers to access (and use) the numerical values and fully evaluate the findings.

- Although statistical associations have been well demonstrated, a short discussion on clinical significance and the impact of this study on patient counselling would strengthen the relevance of the study.

The findings of our study provide insights into the potential for preventive strategies for tinnitus, which are elaborated on in lines 513–520. Regarding patient counselling for individuals with tinnitus, our study examined factors influencing changes in tinnitus impact. However, we currently cannot contextualize these findings in a way that would support individualized counselling or intervention recommendations. As noted in our conclusion, further research is needed to translate these findings into targeted strategies for managing tinnitus impact in affected individuals.

We appreciate the opportunity to revise our manuscript and hope that the changes and clarifications address the concerns raised. Please let us know if any further modifications are required.

Sincerely,

On behalf of all authors,

Denise Fuchten

References

- De Ridder, D., Schlee, W., Vanneste, S., Londero, A., Weisz, N., Kleinjung, T., ... & Langguth, B. (2021). Tinnitus and tinnitus disorder: Theoretical and operational definitions (an international multidisciplinary proposal). Progress in brain research, 260, 1-25.

- Feise, R. J. (2002). Do multiple outcome measures require p-value adjustment?. BMC medical

research methodology, 2(1), 8.

- Hooper, R. (2025). To adjust, or not to adjust, for multiple comparisons. Journal of Clinical Epidemiology, 180, 111688.

- Rothman, K. J. (1990). No adjustments are needed for multiple comparisons. Epidemiology, 1(1), 43-46.

---

## [Editor Report · Decision Letter 1]

29 Dec 2025

Longitudinal analysis of the risk factors for onset and change in tinnitus in the Busselton Healthy Ageing Study

PONE-D-25-16358R1

Dear Dr.Denise Fuchten,

We’re pleased to inform you that your manuscript has been judged scientifically suitable for publication and will be formally accepted for publication once it meets all outstanding technical requirements.

Kind regards,

Gauri Mankekar, MD,PhD,FACS

Academic Editor

PLOS One
---

## [Editor Report · Acceptance letter]

PONE-D-25-16358R1

PLOS One

Dear Dr. Fuchten,

I'm pleased to inform you that your manuscript has been deemed suitable for publication in PLOS One. Congratulations! Your manuscript is now being handed over to our production team.

Kind regards,

on behalf of

Dr. Gauri Mankekar

Academic Editor

PLOS One